# Phosphorus-Mediated Transition from Vegetative to Reproductive Growth in Dwarf Coconut (*Cocos nucifera* L.)

**DOI:** 10.3390/ijms252212040

**Published:** 2024-11-09

**Authors:** Xiaomei Liu, Yi Wu, Mengluo Zhang, Ping Gao, Jing Li, Hao Ding, Xiwei Sun, Lilan Lu, Amjad Iqbal, Yaodong Yang

**Affiliations:** 1Hainan Key Laboratory of Tropical Oil Crops Biology, Hainan Coconut International Joint Research Center, Coconut Research Institute, Chinese Academy of Tropical Agricultural Sciences, Wenchang 571339, China; liuxiaomei@catas.cn (X.L.); wuyi-scuta@163.com (Y.W.); l458455716@163.com (M.Z.); 18293010206@163.com (P.G.); jinghui11111@163.com (J.L.); 17756502642@163.com (H.D.); sunxw065@163.com (X.S.); lilanlu@catas.cn (L.L.); amjadiqbal@awkum.edu.pk (A.I.); 2Department of Food Science & Technology, Abdul Wali Khan University Mardan, Mardan 23200, Pakistan

**Keywords:** *Cocos nucifera*, reproductive growth, phosphorus, *PT1s*, transcriptome

## Abstract

Reducing the time before the flowering stage in coconut (*Cocos nucifera* L.) trees greatly influences yield, yet the mechanisms driving the switch from vegetative to reproductive growth are not well understood, especially the role of phosphorus in this transition. In this study, dwarf coconut plants of the same cultivation age were selected and categorized into the vegetative phase (VP) or the reproductive phase (RP). By examining the phenotypic traits, nutrient variations in the roots and soil, and the transcriptional expression of relevant genes in the roots across both phases, we investigated the potential mechanisms driving the transition from the VP to the RP in coconuts. The shoots of coconuts in the RP were significantly taller compared to those in the VP. Moreover, the phosphorus concentration in the roots of coconuts during the RP was 1.31 times higher than in the VP, which may be linked to the significant upregulation of the *PT1* genes *AZ11G0219160* and *AZ02G0034860* in the roots of coconuts in the RP. In addition, all phosphorus-containing metabolites in the roots during the RP showed a significant increase, particularly those related to long-chain fatty acids and ribonucleotide metabolites. This suggests that coconut roots may facilitate the progression from vegetative to reproductive growth by enhancing phosphorus uptake via *PT1s* and promoting the synthesis and accumulation of phosphorus-containing metabolites.

## 1. Introduction

Coconuts, regarded as a vital tropical oil crop, thrive in Southeast Asia and the Pacific, earning the title of “tree of life” due to its versatility [1]. Its life cycle consists of two main phases: vegetative (foliar) growth and reproductive (sexual) growth. The foliar phase begins with seed germination and ends with floral bud differentiation, lasting 3 to 5 years for early-maturing varieties, like *Cocos nucifera* var. Wenye 3, and up to a decade for late-maturing types, such as *Cocos nucifera* var. Hainan tall. During the RP, inflorescences develop at each leaf axil, leading to the formation of coconut fruits. With new fruit clusters produced every 1 to 2 months, optimizing the progression from foliar to sexual growth has become a crucial area of research to enhance coconut yield [2].

Coconut varieties are categorized into early-blooming and late-blooming types based on the timing of their initial flowering. Early-blooming varieties, primarily dwarf coconuts, are shorter at the flowering onset compared to late-blooming varieties, like tall coconuts, which exhibit more vegetative growth by that time. Research has focused on the genetic modulation of an early bloom in both categories, identifying genes, particularly *SOC1*, *FT*, and *AP1,* in the photoperiodic pathway as crucial for the dwarf coconuts’ flowering [3]. The *FT* gene, which is carried to the apical shoot meristem, coordinates with the bZIP transcription factor *FD* to trigger flowering [4]. Dwarf coconuts show specific genetic variations, including six base pair deletions and alternative splicing in the *FT* gene, which may contribute to their earlier flowering [3]. A linkage map with around 3000 SNPs has been developed from tall and dwarf coconut crosses, leading to the mapping of a significant QTL associated with early-maturation traits and the identification of an *FT* gene analogue [5]. Despite being part of the same variety, dwarf coconuts can still exhibit a 1–3-year difference in initial flowering times, the mechanisms behind which remain elusive.

Floral initiation in plants is an intricate process influenced by ecological variables, in particular light, temperature, and nutrient status, which regulate flowering timing to enhance survival and reproductive success [6,7,8,9]. For instance, under short-day conditions, barley flowers are later to bloom in optimal seasons [10]. Temperature and nutrient levels also play crucial roles: higher temperatures promote rice flowering [11], drought stresses delay flowering in cotton, while *Arabidopsis* exhibits earlier flowering under strong light conditions [12,13]. In addition, high nitrogen levels can postpone flowering whereas low nitrogen levels may trigger earlier flowering [14,15]. Phosphorus serves as a critical macronutrient for promoting plant growth; however, much of the phosphorus in soil is in unavailable organic forms [16]. To acquire phosphorus, plants adapt by modifying root structures and secreting organic acids [17]. Under phosphorus deficiency, they enhance the activity of phosphate transporter genes, like *PHO1* and *PT1s,* to improve phosphorus uptake and recycling [18,19]. In rice, specific transporter genes facilitate phosphorus acquisition and recycling from older to younger tissues [20,21].

This study investigates the progression from the VP to the RP in dwarf coconuts, emphasizing the role of phosphorus. Using molecular biology and bioinformatics techniques, the research examined physiological phenotyping, soil and root nutrient status, gene expression changes, and phosphorus metabolite accumulation during different coconut growth stages. The findings indicated an increased phosphorus uptake capacity in coconut roots during RP. There was also an upregulation of specific phosphate transporter genes (*PT1s*) and a notable rise in phosphorus-containing metabolites. These observations reveal new dimensions of nutrient absorption and highlight potential molecular markers for coconut nutrient management and breeding.

## 2. Results

### 2.1. Phenotypic Analysis of Coconut During VP and RP

To explore the mechanisms driving the transition from the vegetative phase to the reproductive phase in coconuts, we meticulously selected eight coconuts of the same age from a specific region within the same plantation for our study. Of these, four were in the VP and four had progressed to the RP (Figure 1A). It is important to note that coconuts in the RP had formed at least two flower buds, with a maximum of five, while those in the VP showed no signs of flower bud formation (Figure 1B).

In physiological phenotypic analysis of coconut shoots during these two distinct growth phases, we specifically examined the number of mature leaves, an important indicator of biomass. Our results indicated that coconuts in the RP had a significantly greater number of mature leaves than those in the VP (Figure 1C). In addition, since the height and diameter of the stem are vital parameters for assessing the shoot biomass of coconuts, we measured the diameter and height of the main stem across different RPs. However, our measurements showed no significant differences in stem diameter and height among coconuts at various developmental phases despite having the same growth duration (Figure 1D,E).

### 2.2. Nutrient Analysis of Coconut Roots in VP and RP

To determine if the enhancement of coconut reproductive growth is linked to variations in nutrient levels, we carried out a comprehensive nutrient analysis of the roots of coconuts in both the VP and the RP. This analysis specifically targeted the concentrations of phosphorus, sodium, zinc, and boron (Figure 2). The findings showed that the average phosphorus concentration in the roots of coconuts during the VP was 0.68 g/kg, while in those that had transitioned to the RP, the average concentration increased to 0.89 g/kg. This observation indicates a significant increase in phosphorus concentration in the roots of coconuts as they move from vegetative to reproductive growth. However, the concentrations of sodium were 4.87 g/kg in the VP and 4.93 g/kg in the RP, while zinc levels were 39.20 mg/kg in the VP and 44.55 mg/kg in the RP. Boron concentrations were 2.60 mg/kg in the VP and 2.68 mg/kg in the RP. Statistical analysis demonstrated no significant differences in the levels of these elements between the roots of the coconuts at different growth phases.

### 2.3. Soil Nutrient Analysis in Coconut Rhizosphere During VP and RP

To examine how soil nutrient concentrations at various growth stages influence nutrient changes in the roots of coconuts, we performed a nutrient analysis of the soil in the rhizosphere during the VP and RP (Figure 3). The results indicated that the average phosphorus concentration in the soil of coconuts during the VP was 0.26 g/kg, whereas in the RP, it increased to 0.34 g/kg. Additionally, the concentrations of zinc in the soil were 29.93 mg/kg in the VP and 33.55 mg/kg in the RP, while boron concentrations were measured at 6.56 mg/kg in the VP and 8.12 mg/kg in the RP. Although the concentrations of these nutrients were slightly elevated in the soil during the RP compared to the VP, no significant differences were found. However, for sodium, the analysis indicated that the average concentration in the soil during the VP was 2.37 g/kg, while in the soil during the RP, the average concentration increased to 2.84 g/kg. Statistical results indicated a significant difference in sodium concentrations in the soil where coconuts were planted at different growth phases (Figure 3).

### 2.4. Gene Expression Profiling in Coconut Roots During VP and RP

To uncover the molecular mechanisms underlying nutrient absorption changes in coconuts at different reproductive stages, we conducted transcriptome sequencing analysis on the roots of coconuts during the VP and the RP (Figure 4). Initially, we utilized principal component analysis (PCA) to compare the gene expression data from the roots of coconuts in both the VP and the RP, highlighting the differences in gene expression between the two. The analysis revealed that the first (PC1) and second principal components (PC2) collectively explained 54.04% of the total variability, revealing substantial differences in gene expression amongst the coconut roots in the VP and the RP (Figure 4A). Furthermore, we performed a differential gene expression (DEGs) analysis between the VP and RP, identifying a total of 1846 differential genes with a selection threshold of *p* ≤ 0.05. Through heatmap analysis, we observed that the expression profiles of these genes might be widely categorized into two major groups: one group of genes that were upregulated during the RP and another group that was downregulated (Figure 4C). To further analyze these differential genes, we constructed a volcano plot with a selection criterion of *p* ≤ 0.05 and a fold change in expression greater than 1.5. Four phosphate transporter genes that responded to changes were specifically marked on the volcano plot. Preliminary analysis revealed that only the genes *AZ02G0034860* and *AZ11G0219160* exhibited significant differences in expression (Figure 4B). In this research, we conducted a more in-depth analysis of DEGs associated with phosphate metabolism to uncover their roles in the biological processes of coconuts at different reproductive phases (Figure 4D). We conducted a Gene Ontology (GO) enrichment analysis on the 1846 DEGs characterized from the roots of coconuts in both the VP and the RP, specifically concentrating on genes associated with phosphate metabolism. The outcomes of the GO enrichment analysis showed substantial changes in these DEGs across multiple biological processes, especially in the process of inorganic phosphate transmembrane transport.

### 2.5. Role of PT1 Genes in Phosphate Absorption During Reproductive Growth

To further analyze whether phosphate transporter genes are involved in the biological processes of phosphate absorption during RP, we identified and analyzed phosphate transporter genes in coconuts based on the PT1 protein sequences from *Arabidopsis* and rice as models. Through this strategy, we identified seven *PT1* members in the coconut genome (Appendix A). The results of the phylogenetic tree analysis indicated that these coconut *PT1s* have a high degree of sequence similarity with rice *PT1s*, suggesting their potential functional conservation (Appendix A). We further conducted bioinformatics analysis of these seven *PT1s*, including protein sequence length, chromosomal location, protein size, isoelectric point, transmembrane domains, and subcellular localization prediction (Appendix A). These analyses suggest that these genes are likely implicated in the cellular transport of inorganic phosphate. Notably, we found that five of these *PT1s* exhibit expression activity in the roots of coconuts. In particular, genes *AZ11G0219160* and *AZ02G0034860* show significant differential expression between the VP and the RP (Figure 5A). Validation through qPCR further confirmed that *AZ11G0219160* and *AZ02G0034860* have higher expression levels in the roots of coconuts during the RP (Figure 5B). Additionally, we analyzed the tissue expression patterns of these two key *PT1s* in Wenye 3. The results revealed that, with the exception of the pistil, both *AZ11G0219160* and *AZ02G0034860* are expressed in the leaves, stamens, stems, and roots. Among these, *AZ11G0219160* exhibited augmented expression levels in the leaves and stamens, while *AZ02G0034860* showed comparatively elevated expression levels in the leaves, stamens, and roots (Figure 5C,D). Furthermore, we investigated the tissue expression profile of these two genes in another dwarf coconut variety, Wenye 2. The results showed that both genes exhibited exceptionally high expression levels in the leaves (Appendix A). This indicated that these two genes could be key contributors to phosphate uptake and distribution in coconut, which may be related to the high demand for phosphate during the RP of coconut.

### 2.6. Metabolomic Analysis of Phosphorus Compounds in Coconuts During RP

To further explore the synthetic metabolic mechanisms of phosphorus during RP, we conducted a quantitative analysis of phosphorus-containing metabolites in the roots of coconuts at different growth phases. It was found that the levels of 21 phosphorus-containing metabolites significantly increased during RP, predominantly consisting of long-chain fatty acids, including 9 lysophosphatidylethanolamines (LysoPCs) and 9 lysophosphatidylcholines (LysoPEs), 2 ribonucleotide metabolites 2′-Deoxyinosine-5′-monophosphate and nicotinic acid adenine dinucleotide, and 1 alkaloid metabolite O-Phosphocholine (Table 1).

### 2.7. Regulation of Phosphorus Uptake via PT1 Genes in Coconut Development

In this research, we carried out an in-depth analysis of the soil nutrient status around the coconut roots at various growth stages, along with an assessment of nutrient content in the roots and leaves. Based on these analyses, we propose an initial model to explain how coconuts optimize phosphorus absorption and utilization when soil phosphorus levels are sufficient (Figure 6). This process involves regulating the expression of the *PT1* genes to support normal growth and development. Our research findings indicated that when available phosphorus is present in the soil, the expression of *PT1*-associated genes in the roots of coconuts was upregulated. This upregulation might have enhanced the roots’ ability to absorb phosphorus, thereby increasing the phosphorus content in the roots. Additionally, the increase in phosphorus levels in the root promotes the accumulation of phosphorus-containing metabolites in the roots, which in turn plays a positive role in promoting the reproductive growth of coconuts.

## 3. Discussion

### 3.1. Enhancing Coconut Reproductive Growth Through Phosphorus Management

Due to the large biomass of coconuts, we cannot quantify their biomass solely through weight, hence we resort to using stem height, diameter, and leaf count as assessment indicators (Figure 1). The study found that coconuts in the RP have more leaves than those in the VP, increasing biomass, which requires more nutrient support (Figure 1C). The increase in leaf number not only potentially raises the transpiration rate and nutrient absorption of coconuts but also enhances photosynthesis, providing more energy for plant growth, while the development of the root promotes nutrient absorption, particularly phosphorus.

More than 5.7 billion hectares of land across the globe are experiencing a critical shortage of phosphorus that is accessible to plants [22]. Phosphorus is a vital nutrient for vegetative progression and advancement, significantly contributing to important functions like energy transfer, cell division, and signal transduction [23,24]. Insufficient phosphorus supply has a noticeable impact on plant reproductive growth. For instance, in *Arabidopsis*, phosphorus deficiency holds up flowering time [25,26]. In fruit trees, enhancing phosphorus levels can reduce the time it takes from planting to the initial harvest [27,28]. In cereal crops, seeds function as the principal reservoirs of phosphorus, generally representing 60–85% of the total phosphorus found in the plant [24,29]. Therefore, during the RP, plants must absorb sufficient levels of phosphorus to meet the developmental needs of the reproductive organs. In barley, the knockout of the *HvSPDT* gene significantly reduces the transport of phosphorus to the reproductive organs, leading to reduced yield [30]. Our preliminary research results suggest that the absorption efficiency of phosphorus by coconuts may be related to its reproductive growth process. During the reproductive phase (RP) of coconuts, the phosphorus concentration in the roots is significantly higher than in the vegetative phase (VP) (Figure 1 and Figure 2). Although a slight increase in phosphorus concentration in the leaves of coconuts in the RP is not statistically significant, it still implies a potential role for phosphorus in the reproductive growth process of coconuts (Figure 3).

In plants, the *miR399-PHO2* regulatory network performs a central task in maintaining phosphorus homeostasis. Overexpression of *miR399b* and loss of function of *PHO2* lead to the accumulation of excessive phosphorus in plants and result in early flowering [31]. *PSTOL*, which encodes a putative kinase for phosphorus starvation tolerance, promotes plant growth under low-phosphorus conditions. Mutation in the *TaPSTOL* gene results in inefficient phosphorus assimilation and significantly delayed flowering onset [32]. The phosphate transporter *PHOSPHATE1* (*PHO1*) enacts an essential job in the distribution of phosphorus from the roots to above-ground parts. *PHO1* knockout mutants show delayed floral emergence amidst extended and limited daylight regimes [33]. These findings indicate that alterations in gene expression related to phosphorus uptake and translocation impact plant growth and development through modulation of phosphorus acquisition and use. Interestingly, in the roots of coconuts during the RP, the gene expression rates entailed in phosphate transport, *AZ11G0219160* and *AZ02G0034860*, are significantly upregulated (Figure 5B). Bioinformatics analysis revealed that the phosphate transporters encoded by *AZ11G0219160* and *AZ02G0034860* are closely related to the rice phosphate transporters *PT1.4*, *PT1.5*, *PT1.8*, and *PT1.12*, suggesting they may share similar biological functions (Appendix A). In rice, *OsPT1.4* is involved in phosphorus absorption and reactivation processes. Overexpression of *OsPT1;4* significantly increases total plant phosphorus, promotes embryonic development, enhances panicle quality, and increases yield [20,21,34]. *OsPT1.8* is articulated across different tissues, including roots and seeds, and its expression is independent of phosphorus supply. It is responsible for absorbing inorganic phosphorus from the soil and transporting it within the plant [35]. Therefore, we reasonably speculate that the upregulation of coconut *PT1-like* genes may enhance root phosphorus absorption, thereby supporting the phosphorus demands of the coconut’s reproductive growth process. In addition, expression analysis across different tissues revealed that these coconut genes, *AZ11G0219160* and *AZ02G0034860*, have high expression levels in leaves, suggesting they might participate in the transfer and dissemination of phosphorus in leaves (Figure 5C,D). This discovery delivers a unique insight into comprehending the role of phosphorus in coconut reproductive growth and offers potential targets for promoting reproductive growth in coconuts through genetic engineering.

### 3.2. Phosphorus Availability in Soil and Its Impact on Coconut Reproductive Growth Stages

Soil serves as the primary source of nutrients for plants. Studies have indicated that the phosphorus concentration in the soil where coconuts in the RP are planted is slightly higher than that of those in the VP, although this difference is not statistically significant (Figure 3). However, this finding still provides the important insight that in the cultivation process of coconuts, appropriately augmenting the integration of phosphorus fertilizer may promote the absorption of phosphorus by the coconut. This strategy may help to advance the transition of coconuts to the flowering stage more rapidly, thereby enhancing the efficiency of their reproductive growth. In flowering plants, the application of phosphorus fertilizer has been shown to shorten the flowering time of the African marigold by approximately 5–8 days [36]. The increased application of phosphorus fertilizer and inoculation with arbuscular mycorrhizal fungi both effectively enhance the phosphorus absorption efficiency of the roots of snapdragon plants, subsequently increasing the number of floral organs and prolonging the flowering period [37]. The consolidated utilization of phosphorus and zinc has also been proven to affect the growth of floral organs in gladiolus [38].

Soil phosphorus accessibility is governed by a variety of determinants like soil pH, organic matter content, and interactions with other ions. Notably, coconuts demonstrate a robust adaptability to high-salinity conditions, especially thriving in coastal soils. It has been observed that the soil supporting coconut palms during the RP contains significantly higher sodium concentrations compared to that of the VP (Figure 3). However, despite the variation in soil sodium concentration, there is no substantial difference pertaining to sodium level in the roots and leaves of coconuts (Figure 2 and Appendix A). Research indicates that coconuts cultivated for 90 days in 1.23% sodium show a significant increase in phosphorus concentration in the roots. This phenomenon suggests that the coconut’s response to sodium may have a non-specific effect on promoting the absorption of nutrients. Further analysis reveals that under 0.82% sodium treatment, the absorption of potassium, calcium, magnesium, and nitrogen by coconuts is also significantly enhanced [39]. Based on these findings, we speculate that the concentration of sodium in the soil may regulate the efficiency of phosphorus absorption by the roots, thereby affecting the transition of coconut palms from the VP to the RP. Considering that the impact of soil nutrient status on the nutrient content in the roots of coconuts is an intricate process encompassing the interplay of multiple factors, additional research is needed to better understand the specific mechanisms at play.

### 3.3. The Role of Phospholipids in the RP of Coconut

A significant accumulation of phosphorus-containing metabolites was found in the roots of coconuts in the RP, including lysophosphatidylethanolamine (LysoPC) and lysophosphatidylcholine (LysoPE), among others (Table 1). The accumulation of phospholipid metabolites may provide a necessary phosphorus source for the growth of coconut floral buds [40]. In maize, the High *PhosphatidylCholine 1* (*HPC1*) is responsible for encoding a phospholipase A1 enzyme which regulates the flowering time by modulating lysophosphatidylcholine and is involved in the adaptation process to the cold environment at high altitudes [41]. Interestingly, in addition to lysophosphatidylcholine, the phospholipid phosphatidylglycerol (PG) in the cell membrane can interact with florigen and integrate into the plasma membrane bilayer, regulating the floral development in plants [4]. Furthermore, in rice, it has been found that the hydrolysis process of lysophosphatidylcholine mediated by phospholipase D significantly inhibits the heading process of rice [42]. This is similar to the observations in this study, where an increase in phospholipid levels in coconuts is clearly correlated with flowering [43]. Additionally, the accumulation of phosphorus-containing metabolites may be related to the upregulation of phosphate transporter genes [44,45]. Therefore, we infer that during the reproductive growth phase, coconuts enhance absorption of phosphorus and increase the synthesis of metabolites such as phospholipids, promoting their transition to the RP.

### 3.4. The Potential Role of Plant Hormones in the RP of Coconuts

In fact, the reproductive transition in plants is a complex and profound process, in which plant hormones typically play a significant role [46,47,48]. In coconuts, we have observed significant differences in genes associated with plant hormones across various growth stages (Appendix A). Compared to the VP, the *AZ08G0166160* gene encoding the GA2OX3 protein, is markedly downregulated in the root during the RP. The GA2OX3 protein functions in the degradation and metabolism of gibberellins, primarily responsible for regulating the deactivation of biologically active gibberellins. Interestingly, related studies have shown that phosphorus starvation in plants significantly suppresses the levels of active GA in the root, and GA has a significant stimulating effect on plant growth and flowering [49,50]. In addition to genes related to gibberellin metabolism, the *AZ03G0063350* gene encoding the CKX5 protein, associated with cytokinin metabolism, also exhibits significant downregulation in expression during the RP. Furthermore, we have found that key genes involved in auxin transport, PINs and LAXs, as well as abscisic acid receptors PYR/PYLs, ethylene receptor ETR2, and brassinosteroid LRR receptor kinase BRL2, are all significantly downregulated during the RP. This suggests that plant hormones may be involved in the growth and development process of coconuts. However, it is worth exploring whether the expression changes of these plant hormone-related genes have a direct regulatory relationship with phosphorus levels in the root, and further in-depth research is needed to prove this.

This study solely analyzed and discussed the changes in nutrient status and gene expression, as well as how these variations impact the reproductive growth of coconuts. It is hoped that these findings will provide new research perspectives for understanding the reproductive physiology of coconuts and offer potential strategies for enhancing their reproductive capacity through nutritional management and genetic engineering.

## 4. Materials and Methods

### 4.1. Coconut Growth

This study examines the *Cocos nucifera* var. Wenye 3 as the research subject, a new variety that originated from the Malaysian red dwarf species and has been domesticated by humans. This variety begins to enter the reproductive phase at around 5–6 years. The experimental site was located at a coconut plantation in Tunchang County, Hainan Province, China. During the experiment, the coconut palms were subjected to uniform light exposure and irrigation management. The climatic characteristics of the area include an annual mean high temperature of 30 °C and a mean low temperature of 21 °C. All phenotypic data of the coconuts were collected from within the same plot. The soil in this coconut plantation is typical of the acidic soils in southern China, with a pH of 4.5.

### 4.2. Ion Quantification in Coconuts

This study conducted ion concentration assays on both the new roots and functional leaves of coconuts. The collected samples were washed three times with distilled water and then underwent a final rinse with ultrapure water. The coconut plant samples were subjected to high-temperature drying by blanching at 105 °C for 30 min and drying at 70 °C to a constant weight, followed by being grinded into a powder for room-temperature storage. The soil samples were collected, air-dried in the shade, mixed, ground, sifted through a 100-mesh sieve, and stored at room temperature. The samples were powdered, and 0.5 g of the resulting powder was placed into a glass tube for digestion. The mixture was then gradually heated in 6 mL of concentrated nitric acid to 130 °C and digested for 1 h. Next, 1–2 mL of hydrogen peroxide was added to clarify the solution and digestion continued for an additional 30 min. After cooling, the solution was adjusted to a final volume of 25 mL to ready the sample solution for analysis. Soil samples weighing 0.2 g were placed into polytetrafluoroethylene (PTFE) digestion tubes for digestion. To these samples, 6 mL of nitric acid, 1 mL of hydrochloric acid, and 3 mL of hydrofluoric acid were added, after which the mixture was digested in a microwave digestion system. Nutrient measurements were conducted using an Agilent 7700x Inductively Coupled Plasma Mass Spectrometer (ICP-MS) (Agilent, Santa Clara, CA, USA) to quantify elements, including phosphorus, sodium, zinc, and boron.

### 4.3. Preparation and Sequencing of Coconut Transcriptomes

The experiment focused on Wenye 3, with new roots collected from three plants in both the vegetative and reproductive phases for transcriptome sequencing. The samples were promptly rinsed with distilled water to eliminate surface soil, then blotted dry and rapidly frozen in liquid nitrogen before being transported on dry ice to Metware Biotechnology Co., Ltd. in Wuhan, China for transcriptome sequencing. In this study, a rigorous low-temperature processing protocol was followed for sequencing plant samples. After detection, the samples underwent low-temperature drying treatments and were eventually returned using dry ice. The procedure involved extracting RNA from coconut root systems, preparing RNA libraries, sequencing, filtering data, aligning the sequences to the dwarf coconut genome, calculating gene expression FPKM values, and annotating gene functions. Using the gene expression FPKM values, genes expressed in both the vegetative and reproductive phases of coconuts were analyzed and differentially expressed genes were identified with a threshold of *padj* lesser than 0.05 and an absolute log2FC (fold change) greater than 0.5.

### 4.4. Metabolite Profiling

The materials utilized for metabolite detection were the same as those used for transcriptome sequencing, with a portion of the root samples designated for transcriptome analysis also being set aside for metabolite profiling. We placed the coconut samples into the freeze dryer (Scientz-100F, Ningbo, China) and then used a grinder to pulverize the samples into powder (frequency 30 Hz, 1.5 min). Next, we weighed 50 milligrams of the sample powder and added 1.2 mL of pre-cooled 70% methanol water extraction solution. We vortexed every 30 min for 30 s for a total of 6 times. After centrifugation (12,000 rpm, 3 min), we collected the supernatant, filtered the sample through a microporous membrane (pore size 0.22 μm), and stored it in an injection vial for UPLC-MS/MS analysis. The metabolites were then analyzed using a liquid chromatography column in conjunction with an ultra-performance liquid chromatography-electrospray ionization-tandem mass spectrometry (UPLC-ESI-MS/MS) system (Agilent, USA). The mobile phase was composed of a 0.1% aqueous solution of formic acid and a 0.1% acetonitrile solution of formic acid with a flow rate of 0.35 mL/min and a column temperature set at approximately 40 °C.

### 4.5. Expression Analysis of PT1 Genes

To validate the expression changes of key *PT1* genes, roots were collected from coconut varieties Wenye 3 and Wenye 2 during both the vegetative and reproductive phases. The total RNA was then extracted using the RNA Extraction Kit. Following this, 250 ng of total RNA was reverse-transcribed into cDNA with the All-in-One Gold One-Step cDNA Synthesis Kit and diluted 10-fold for qPCR to assess the expression changes of the target genes. The qPCR cycling parameters included an initial denaturation step at 95 °C for 1 min, followed by 40 cycles comprising denaturation at 95 °C for 15 s, annealing at 60 °C for 15 s, and extension at 72 °C for 30 s. The coconut *Actin* gene served as the endogenous control, and the relative levels of gene expression were determined using the ΔΔCt method (Appendix A).

### 4.6. Protein Sequence Alignment and Phylogenetic Analysis of PT1 Proteins

Protein sequences for the PT1 family members from rice, *Arabidopsis*, and the dwarf coconut PT1 were retrieved from online databases. Subsequently, a phylogenetic tree was constructed using the neighbor-joining method along with the Poisson model. The protein sequence data for rice PT1 were retrieved from the MSU-RGAP (https://rice.uga.edu/, accessed on 5 November 2024), while the PT1 protein sequences for *Arabidopsis* were acquired from the TAIR (https://www.arabidopsis.org/, accessed on 5 November 2024). The phylogenetic tree was generated using MEGA (version 10.2.6), and the conserved structures of the protein sequences were examined alongside Jalview.

### 4.7. Statistical Analysis

The information was presented as the average value ± standard error (SE) obtained from three separate repetitions. The statistics were performed using SPSS software (version 19.0; SPSS, Chicago, IL, USA). Student’s *t*-tests were applied to ascertain the significance of differences when *p* < 0.05. A significance level of *p* < 0.05 was considered statistically meaningful.

## 5. Conclusions

This study has unveiled the potential role of phosphorus in the transition of coconuts from the VP to the RP. The findings suggest that coconuts in the RP exhibit a greater number of leaves, indicating a significant increase in biomass. Moreover, a notable rise in phosphorus concentration within the roots of coconuts during the RP is closely correlated with their transition to reproductive growth. Gene expression analysis reveals that *PT1* genes, which are linked to phosphorus uptake, are upregulated in the RP, potentially boosting the roots’ phosphorus absorption capacity. Additionally, the accumulation of phospholipid metabolites in the roots of coconuts during the RP offers an essential phosphorus resource for the development of floral buds. However, the reproductive transition in plants is a complex process, where plant hormones may exert a significant influence. Future research will concentrate on clarifying the regulatory mechanisms of phosphorus on hormones to achieve a deeper comprehension of its potential role in the reproductive transition of coconuts.

## Figures and Tables

**Figure 1 ijms-25-12040-f001:**
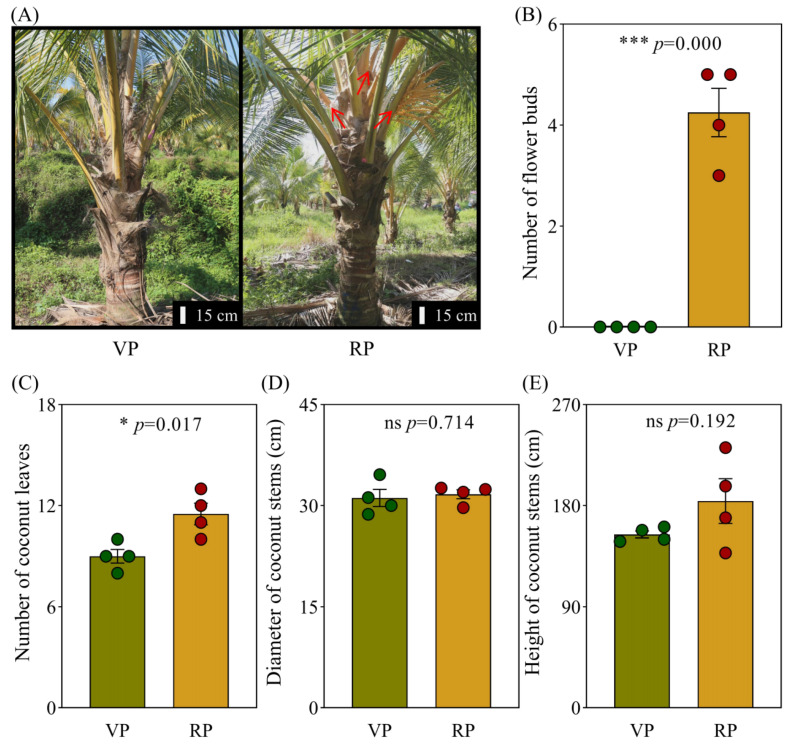
Physiological phenotypic characteristics of coconut in different growth phases (**A**) Six-year-old dwarf coconut in the vegetative and reproductive phases. (**B**) Number of floral buds in dwarf coconut at different growth phases. (**C**) Number of fully expanded leaves in dwarf coconut at different growth phases. (**D**) Stem width of dwarf coconut at different growth phases. (**E**) Stem height of dwarf coconut at different growth phases. VP, vegetative phase; RP, reproductive phase. The red arrow points to the coconut flower. Each treatment was performed with four biological repetitions, the dots represent the biological replicates, with green representing VP coconuts and red representing RP coconuts. Asterisks point to statistically significant differences; ns depicts insignificance; * depicts significant difference at the 0.01 < *p* ≤ 0.05 level; *** depicts significant difference at the *p* ≤ 0.001 level.

**Figure 2 ijms-25-12040-f002:**
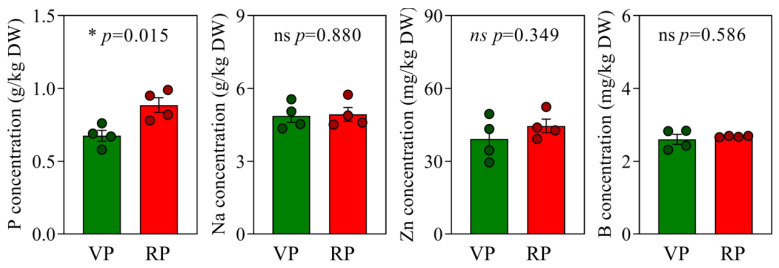
Nutrient status of new roots in coconuts at different growth phases. VP, vegetative phase; RP, reproductive phase. Each treatment was performed with four biological repetitions, the dots represent the biological replicates, with green representing VP coconuts and red representing RP coconuts. Asterisks point to statistically significant differences; ns depicts insignificance; * depicts significant difference at the 0.01 < *p* ≤ 0.05 level.

**Figure 3 ijms-25-12040-f003:**
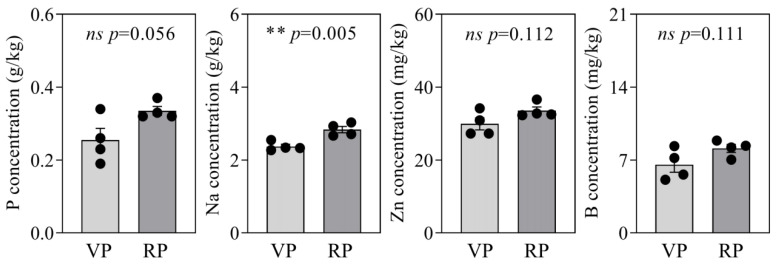
Analysis of nutrient status in the soil of coconuts at different phases. VP, vegetative phase; RP, reproductive phase. Each treatment was performed with four biological repetitions, the dots represent the biological replicates. Asterisks point to statistically significant differences; ns depicts insignificance; ** depicts significant difference at the 0.001 < *p* ≤ 0.01 level.

**Figure 4 ijms-25-12040-f004:**
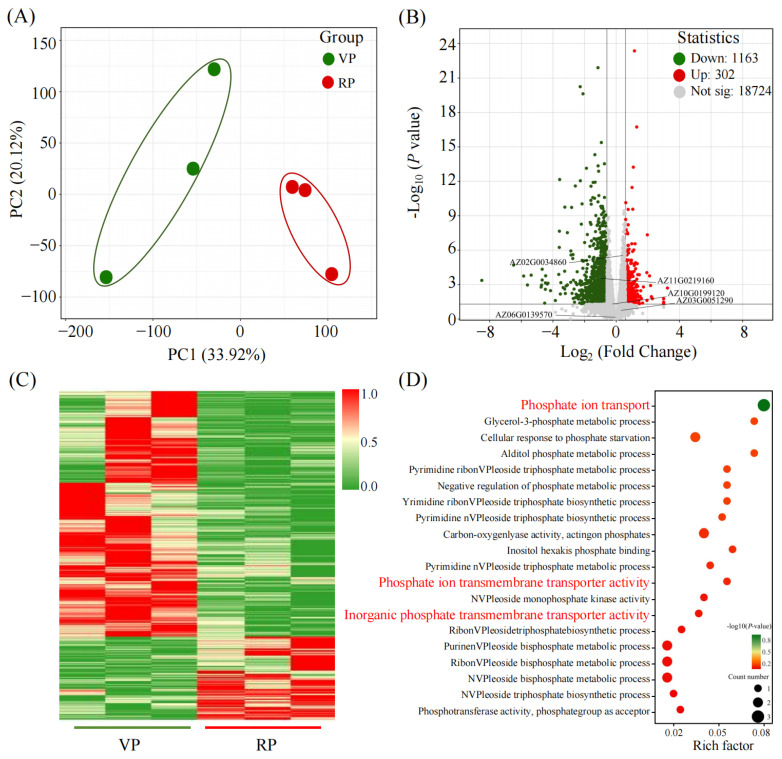
Changes in gene transcriptional expression in the roots of coconuts at different phases. (**A**) Principal component analysis (PCA) of DEGs in the roots of coconuts. (**B**) Volcano plot analysis of DEGs in the roots of coconuts. (**C**) Heatmap analysis of DEGs in the roots of coconuts. (**D**) Perform Gene Ontology (GO) enrichment analysis on phosphorus-related DEGs to determine the most significant enrichment pathway at the threshold *p* ≤ 0.05, The red font represents the enrichment to gene entries related to phosphorus transport. Each treatment was conducted with three biological repetitions.

**Figure 5 ijms-25-12040-f005:**
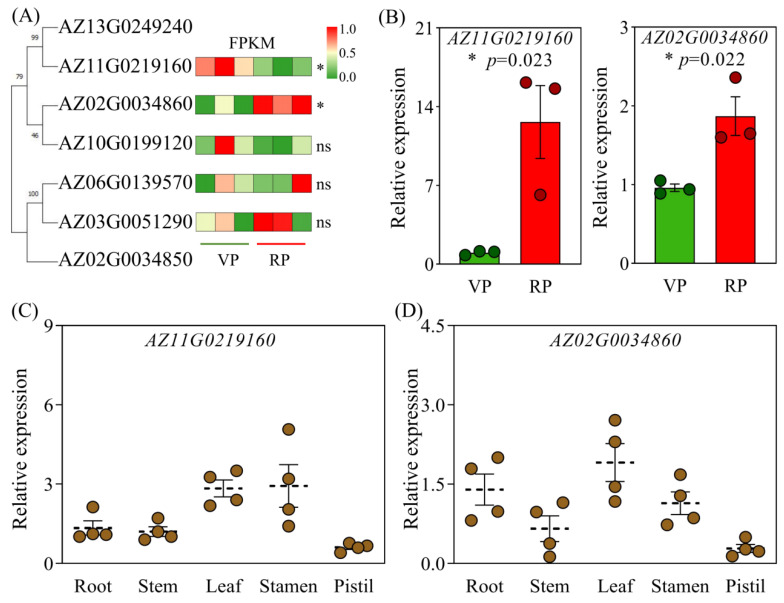
Expression patterns of *PT1* genes in coconuts at different phases. (**A**) Phylogenetic analysis of the *PT1* family of phosphate transporter genes in coconuts and of gene transcriPT1onal expression changes across different growth phases. (**B**) qPCR analysis of two *PT1s* genes in the roots of coconuts at various reproductive Phases, the dots represent the biological replicates, with green representing VP coconuts and red representing RP coconuts. (**C**) Expression levels of the coconut phosphate transporter gene *AZ11G0219160* in the roots, stems, leaves, stamen and pistil of Wenye 3 during the reproductive phases. (**D**) Expression levels of the coconut phosphate transporter gene *AZ02G0034860* in the roots, stems, leaves, stamen and pistil of Wenye 3 during the reproductive phases. VP, Vegetative phase; RP, Reproductive phase. Each treatment was performed with four biological repetitions, the dots represent the biological replicates. Asterisks point to statistically significant differences; ns depicts insignificance; * depicts significant difference at the 0.01 < *p* ≤ 0.05 level.

**Figure 6 ijms-25-12040-f006:**
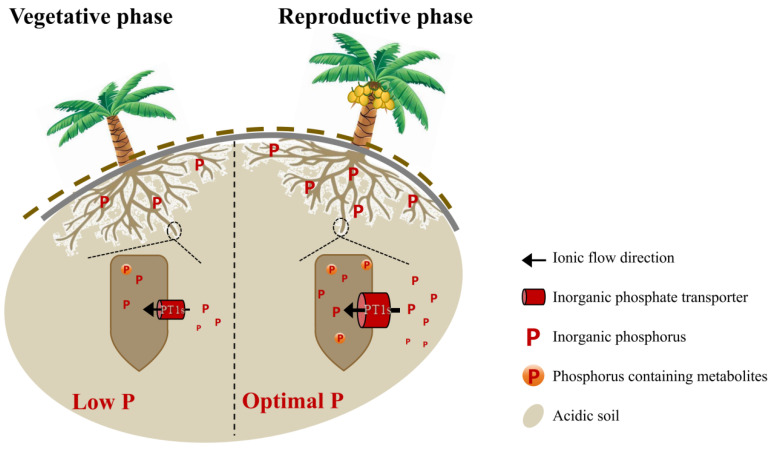
Schematic diagram of phosphorus absorption patterns in coconuts at different phases. Coconuts primarily grow in acidic soils, and in the presence of adequate phosphorus, they can enhance the absorption of inorganic phosphorus by upregulating the expression of *PT1* genes which promote the reproductive growth process of the coconut. Under conditions of severe phosphorus deficiency, however, the expression of *PT1* genes in the roots of coconuts significantly decreases, and the phosphorus absorption process they mediate is substantially reduced. This leads to a significant inhibition of the production and buildup of phosphorus-containing metabolites in the roots, thereby affecting the reproductive growth of the coconut.

**Table 1 ijms-25-12040-t001:** Variations in phosphorus-containing metabolites in the roots of coconuts at different phases.

NO.	Compounds	Formula	Class	UC Peak Area × 10^4^ (Means ± Standard Error)	FC Peak Area × 10^4^ (Means ± Standard Error)	FC/UC
1	LysoPE 18:4	C_23_H_40_NO_7_P	Lipids	7.08 ± 2.86	13.41 ± 5.32	2.77
2	LysoPE 15:1	C_20_H_40_NO_7_P	Lipids	24.79 ± 11.15	43.69 ± 5.85	1.49
3	LysoPE 20:2	C_25_H_48_NO_7_P	Lipids	39.81 ± 12.09	59.45 ± 8.25	1.46
4	LysoPE 20:5	C_25_H_42_NO_7_P	Lipids	147.83 ± 61.75	228.44 ± 71.18	1.87
5	LysoPE 20:4	C_25_H_44_NO_7_P	Lipids	903.89 ± 177.71	1354.33 ± 274.01	1.83
6	LysoPE 20:3	C_25_H_46_NO_7_P	Lipids	1070.13 ± 382.39	1806.95 ± 590.46	1.92
7	LysoPE 20:2(2n isomer)	C_25_H_48_NO_7_P	Lipids	23.79 ± 5.58	48.05 ± 14.84	1.79
8	LysoPE 20:4(2n isomer)	C_25_H_44_NO_7_P	Lipids	890.70 ± 178.09	1356.83 ± 262.55	1.79
9	LysoPE 20:3(2n isomer)	C_25_H_46_NO_7_P	Lipids	1038.17 ± 343.54	1792.80 ± 568.66	2.01
10	LysoPC 22:5	C_30_H_52_NO_7_P	Lipids	9.07 ± 1.27	12.99 ± 1.00	1.37
11	LysoPC 22:4	C_30_H_54_NO_7_P	Lipids	30.57 ± 5.63	53.06 ± 8.80	2.11
12	LysoPC 20:2	C_28_H_54_NO_7_P	Lipids	44.10 ± 9.42	70.38 ± 12.19	1.68
13	LysoPC 18:4	C_26_H_46_NO_7_P	Lipids	59.84 ± 26.05	103.81 ± 44.18	3.53
14	LysoPC 20:5	C_28_H_48_NO_7_P	Lipids	1227.68 ± 421.16	1742.49 ± 336.07	1.81
15	LysoPC 20:3	C_28_H_52_NO_7_P	Lipids	1975.87 ± 592.86	3572.95 ± 1177.37	2.19
16	LysoPC 20:4	C_28_H_50_NO_7_P	Lipids	2364.75 ± 471.76	3626.17 ± 791.73	2.07
17	LysoPC 16:2(2n isomer)	C_24_H_46_NO_7_P	Lipids	18.27 ± 6.43	30.18 ± 9.65	2.41
18	LysoPC 20:2(2n isomer)	C_28_H_54_NO_7_P	Lipids	54.51 ± 19.44	83.05 ± 21.39	1.68
19	Nicotinic acid adenine dinucleotide	C_21_H_27_N_7_O_14_P_2_	Nucleotides and derivatives	11.17 ± 0.00	71.81 ± 3.99	6.68
20	2′-Deoxyinosine-5′-monophosphate	C_10_H_13_N_4_O_7_P	Nucleotides and derivatives	29.09 ± 15.29	57.36 ± 17.49	1.86
21	O-Phosphocholine	C_5_H_15_NO_4_P	Alkaloids	278.48 ± 13.49	469.65 ± 50.64	1.58

## Data Availability

The original contributions presented in the study are included in the article/Appendix A, further inquiries can be directed to the corresponding author.

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
