# Peer review of "Phosphorus-Mediated Transition from Vegetative to Reproductive Growth in Dwarf Coconut (*Cocos nucifera* L.)"

_ijms, 2024, doi:10.3390/ijms252212040_

Round 1

Reviewer 1 Report

Comments and Suggestions for Authors

The paper is original in the design, and focus on the transition of the vegetative to the reproductive stage in a tropical plant of economic interest. Although the expreimental design and the results seem correct, the major problem with this manuscript is the interpretation of the results. Authors interpret that all the changes depend on phosphate absorption and transport. This mechanism is quite simple, and authors are not considering alternative hypothesis or other parts of the system. From this perspective is quite strange that in figure 4 authors only consider in the transcriptome analysis genes related to phosphate metabolism. The transition to the vegetative to the reproductive stage is an hormonal driven mechanisms that depends on environmental signals. If the hypothesis is true, and all depends on the phosphate absorption, the signal should be transmitted from the root to the aerial part somehow. The main hormones participating in this physiological process are gibberelins. So:

How are the genes related to gibberelein biosynthesis and degradation in the transcriptome experiment?

Is this change dependent on a gibberelin peak or are other hormones participating?

I recommend to analyze the genes related to hormonal regulation in the transcriptome analysys and quantify the hormonal levels in vegetative or reproductive stage plants, in order to complete the model. Knoing if this a gibberellin dependent or independent process is crucial for the understanding.

Author Response

Dear Reviewer,

Thank you for your valuable feedback on our manuscript. We have carefully considered your comments and have made the necessary revisions to address each point. Below are our responses to your specific concerns:

Comments 1: How are the genes related to gibberelein biosynthesis and degradation in the transcriptome experiment?

Response 1: Thank you for pointing this out. Therefore, we examined all genes related to the synthesis and metabolism of plant hormones. By screening for differential genes (p < 0.05, Log2FC ≥ 0.5), we found that most genes involved in gibberellin synthesis and metabolism did not show significant expression changes. We only identified the gene AZ08G0166160, which is involved in gibberellin degradation, to be significantly downregulated during the reproductive growth stage of coconuts (Fig. S5 A). This gene encodes the gibberellin 2 beta-dioxygenase enzyme G2OX3, which plays a role in the degradation of gibberellins, potentially affecting the levels of gibberellins in the roots of coconut. (Page 8-9, Line 304-315)

Comments 2: Is this change dependent on a gibberelin peak or are other hormones participating?

Response 2: In addition to gibberellin metabolism-related genes, we also examined genes related to other plant hormones, including auxin, ethylene, abscisic acid, salicylic acid, cytokinins, and jasmonic acid. Besides the gibberellin metabolism-related gene G2OX3, we found that the cytokinin catabolic gene AZ03G0063350, which encodes cytokinin dehydrogenase 5 (CKX5), was significantly downregulated during the reproductive phase of coconuts (Fig. S5 B). In addition to this, we observed that genes involved in the encoding of auxin transport proteins PINs and LAXs (Figs. S6, S7), as well as genes related to the signaling of ethylene receptor proteins and the encoding of brassinosteroid LRR receptor kinase BRL2 (Fig. S8), were all significantly downregulated in the root of coconuts during their reproductive phase. This suggests that, in addition to gibberellins and cytokinins, auxin, ethylene, and brassinosteroids may be involved in the reproductive development process of coconuts.(Page 9, Line 315-321)

Comments 3: I recommend to analyze the genes related to hormonal regulation in the transcriptome analysys and quantify the hormonal levels in vegetative or reproductive stage plants, in order to complete the model. Knoing if this a gibberellin dependent or independent process is crucial for the understanding.

Response 3: Through the analysis of gene expression changes, we observed significant alterations in the expression of genes related to plant hormones across different growth stages. This led us to consider whether nutrient changes in coconuts might affect reproductive transitions by regulating changes in plant hormones. As coconuts transition from the vegetative to the reproductive phase, they undergo a lengthy period of vegetative growth, resulting in their typically tall stature. Consequently, nutrients and metabolites in the coconut root system must travel a considerable distance to reach the above-ground tissues, where they regulate the growth and development of the coconut palm. Therefore, we believe that directly examining the above-ground tissues could be an appropriate material for uncovering the reproductive shift in coconuts. However, we currently lack materials from the apical tissues at different growth stages, making it difficult to directly analyze changes in plant hormones within the apical meristem to assess their role in the reproductive process. We plan to include this in future studies. We agree that quantifying hormone levels will help gain a deeper understanding of the regulatory mechanisms underlying the transition from vegetative to reproductive growth in coconuts. (Page 11, Line 426-429)

We appreciate your valuable comments, which not only help us expand our thinking on subsequent studies of coconut growth stage transitions but also provide new perspectives for our research.

Sincerely,

Dr. Xiaomei Liu

Reviewer 2 Report

Comments and Suggestions for Authors

The manuscript by Liu et al examines how phosphorus aids the transition from vegetative to reproductive growth in coconut. Dwarf coconuts in the reproductive phase had higher root phosphorus levels and increased expression of PT1 genes compared to those in the vegetative phase. Elevated phosphorus-containing metabolites, especially in fatty acids and ribonucleotides, suggest that enhanced phosphorus uptake and metabolism may facilitate flowering in coconut. The topic is very important and is appropriate to this journal. However, the manuscript suffers from some minor problems. The manuscript should be checked in terms of English grammar correction. 

1. Information about the soil type, coconut variety’s genetic background, or rootstock used could enhance reproducibility.

2. How were samples stored after drying to prevent potential contamination or degradation?

3. The method for metabolite profiling could be modified with additional details on the lyophilization and methanol extraction process to improve reproducibility.

4. What might account for the discrepancy in leaf number without changes in stem diameter or height, and could this impact phosphorus dynamics?

5. Could any external environmental factors (e.g., seasonal variation) have influenced nutrient availability in the soil or coconut roots?

6. Although phosphorus levels increased, the impact of sodium concentration on reproductive growth remains speculative. Adding a discussion on potential physiological effects of soil sodium would be beneficial.

7. Is the RNA-seq data publicly available (e.g., in the NCBI SRA or GEO database) for transparency and independent validation? 

8. The conlusion section needs to be provide to present an overview of the core findings from this work and emphasize the future aspects of this work.

9. The whole MS should be checked thoroughly for gammar check and typos. 

Author Response

Dear Reviewer,

Thank you for your valuable feedback on our manuscript. We have carefully considered your comments and have made the necessary revisions to address each point. Below are our responses to your specific concerns:

Comments 1: Information about the soil type, coconut variety’s genetic background, or rootstock used could enhance reproducibility.

Response 1: Thank you for pointing this out. Therefore, we have expanded the methods section to include additional details on the soil characteristics of this coconut plantation, which are indicative of the acidic soils prevalent in southern China, featuring a pH of 4.5. The study focuses on Cocos nucifera var. Wenye 3, a new variety derived from the Malaysian red dwarf species and domesticated by humans, which begins reproductive phase around 5-6 years. (Page 9, Line 329-331; Line 336-337)

Comments 2: How were samples stored after drying to prevent potential contamination or degradation?

Response 2: We are pleased to have your expert input on this matter. We have addressed the deficiencies in the material handling information within the manuscript. For all experimental samples, there were two storage methods used. Samples for nutrient analysis of coconut tissues and soil underwent high-temperature drying. The collected plant tissues were brought back to the laboratory, blanched at 105℃ for 30 minutes, and then dried at 70℃ until a constant weight was achieved. After drying, the samples were ground and stored at room temperature. Soil samples were collected from the field and allowed to dry completely in a cool place. After drying, the soil was mixed, and a portion was ground and stored at room temperature. For molecular sequencing of plant samples, low-temperature handling was maintained throughout the process. After collection, samples were rapidly frozen in liquid nitrogen and then transported on dry ice to the testing company. The company handled the samples at low temperatures and finally returned the freeze-dried samples using dry ice. (Page 9, Line 342-346; Page 10, Line 364-366)

Comments 3: The method for metabolite profiling could be modified with additional details on the lyophilization and methanol extraction process to improve reproducibility.

Response 3: Many thanks for highlighting this issue. We have expanded the methods section to include more details on freeze-drying and methanol extraction processes to enhance repeatability. The coconut samples were subjected to freeze-drying using a Scientz-100F freeze dryer, followed by grinding into a fine powder with a grinder operating at a frequency of 30 Hz for 1.5 minutes. Subsequently, 50 mg of the resultant powder was precisely weighed and combined with 1.2 mL of a pre-cooled 70% methanol extraction solution. The mixture underwent vortexing at 30-second intervals over a period of 30 minutes, repeated six times. Following centrifugation at 12,000 rpm for 3 minutes, the supernatant was carefully collected and filtered through a 0.22-micron microporous membrane before being transferred to injection vials for subsequent analysis via UPLC-MS/MS. The metabolites underwent analysis via a liquid chromatography column coupled with an ultra-high-performance liquid chromatography-electrospray ionization-tandem mass spectrometry system. The mobile phase was composed of a 0.1% formic acid aqueous solution and a 0.1% formic acid acetonitrile solution, with a flow rate of 0.35 mL/min and the column temperature was set at approximately 40℃. (Page 10, Line 377-383)

Comments 4: What might account for the discrepancy in leaf number without changes in stem diameter or height, and could this impact phosphorus dynamics?

Response 4: Due to their large biomass, coconut cannot be quantitatively analyzed through traditional weighing methods. We used stem height, diameter, and leaf count as indicators to measure the aboveground biomass of coconut. The analysis revealed that stem dimensions did not significantly differ across various growth stages, while biomass changes were primarily reflected in leaf numbers. Coconut in the reproductive phase have a greater biomass than those in the vegetative phase, requiring more nutrients. An increase in leaf number may enhance transpiration, strengthening water and nutrient absorption, potentially affecting phosphorus uptake by the roots. Additionally, more leaves mean more photosynthesis, providing more photosynthetic products for root and other tissue growth, while root growth, in turn, aids in nutrient absorption, including phosphorus.(Page 4, Line 196-202)

Comments 5: Could any external environmental factors (e.g., seasonal variation) have influenced nutrient availability in the soil or coconut roots?

Response 5: Coconut are tropical, woody plants that demand stringent conditions regarding temperature and light, and are sensitive to seasonal changes. Particularly during winter, these fluctuations can significantly hinder vital processes such as photosynthesis. Lu's research on coconuts under cooler conditions has revealed that lower temperatures can significantly inhibit photosynthesis in coconut leaves and impede the growth of new leaf (Lu et al., 2023). This physiological reaction to cold conditions may also manifest in the root system, where cooler temperatures could potentially hinder nutrient uptake by the coconut's roots. However, this hypothesis necessitates further experimentation to substantiate the conjecture.

Comments 6: Although phosphorus levels increased, the impact of sodium concentration on reproductive growth remains speculative. Adding a discussion on potential physiological effects of soil sodium would be beneficial.

Response 6: We are pleased to have your expert input on this matter. Although the effect of sodium concentrations on reproductive growth is still a matter of speculation, we have added a discussion on the potential physiological impacts of soil sodium, drawing on recent research that suggests a non-specific influence on nutrient uptake (Tang and Li, 2024).(Page 6-7, Line 267-270; Line 272-279)

Comments 7: Is the RNA-seq data publicly available (e.g., in the NCBI SRA or GEO database) for transparency and independent validation?

Response 7: Due to the data also forming part of the lead author's thesis, we are unable to upload the raw data to public databases at this time. However, we are open to sharing our data upon request to facilitate transparency and independent verification.

Comments 8: The conlusion section needs to be provide to present an overview of the core findings from this work and emphasize the future aspects of this work.

Response 8: We appreciate your vigilance on this matter. We have added a conclusion section that outlines the core findings of our work and emphasizes the future aspects of this research. (Page 11, Line 418-429)

Comments 9: The whole MS should be checked thoroughly for gammar check and typos.

Response 9: Thank you for pointing this out. The manuscript has been thoroughly reviewed for grammar and spelling errors by a native English-speaking researcher, Dr. Amjad Iqbal, ensuring that the language is clear and accessible to readers.

We have carefully addressed each of your suggestions and believe that the revisions have strengthened our manuscript. We look forward to your further guidance and hope our changes meet your expectations.

Sincerely,

Dr. Xiaomei Liu

References

Lu, L.; Yang, W., Dong, Z.; Tang, L.; Liu, Y.; Xie, S.; Yang, Y. Integrated transcriptomic and metabolomics analyses reveal molecular responses to cold stress in coconut (Cocos nucifera L.) seedlings. International Journal of Molecular Sciences. 2023, 24 (19), 14563-14563.

Tang, L.; Li, Y. Effects of sodium chloride application on major mineral nutrient elements in leaves of coconut seedlings. Tropical Agricultural Science & Technology. 2024, 47 (3), 16-20.

Round 2

Reviewer 1 Report

Comments and Suggestions for Authors

Authors have greatly improved the manuscript. I can recommend publication.